# Statin Use and Long-Term Mortality after Rectal Cancer Surgery

**DOI:** 10.3390/cancers13174288

**Published:** 2021-08-25

**Authors:** Arvid Pourlotfi, Gary Alan Bass, Rebecka Ahl Hulme, Maximilian Peter Forssten, Gabriel Sjolin, Yang Cao, Peter Matthiessen, Shahin Mohseni

**Affiliations:** 1Department of Surgery, Orebro University Hospital, 701 85 Orebro, Sweden; maximilian.forssten@oru.se (M.P.F.); gabriel.sjolin@regionorebrolan.se (G.S.); peter.matthiessen@regionorebrolan.se (P.M.); 2School of Medical Sciences, Orebro University, 702 81 Orebro, Sweden; gary.bass@pennmedicine.upenn.edu (G.A.B.); rebecka.ahl-hulme@sll.se (R.A.H.); 3Division of Traumatology, Emergency Surgery & Surgical Critical Care, University of Pennsylvania, Philadelphia, PA 19104, USA; 4Division of Trauma and Emergency Surgery, Department of Surgery, Karolinska University Hospital, 171 76 Stockholm, Sweden; 5Division of Surgery, Department of Clinical Science, Intervention and Technology, Karolinska Institutet, 141 52 Stockholm, Sweden; 6Clinical Epidemiology and Biostatistics, School of Medical Sciences, Orebro University, 701 82 Orebro, Sweden; yang.cao@oru.se; 7Division of Trauma and Emergency Surgery, Department of Surgery, Orebro University Hospital, 701 85 Orebro, Sweden

**Keywords:** rectal cancer, oncological rectal surgery, statin therapy, mortality

## Abstract

**Simple Summary:**

A diagnosis of locally advanced rectal cancer and succeeding surgery remains an area of high postoperative risk for adverse outcomes. The current investigation aims to clarify uncertainty regarding the impact of ongoing statin therapy on postoperative long-term mortality rates after curative surgical resections of rectal cancer by examining data from a large validated national register. It is the first to date to investigate the impact of statin therapy on long-term mortality following curative rectal cancer surgery. Having an ongoing statin prescription was associated with a lower risk of mortality up to five years after surgery. The results should be confirmed in future large, randomized clinical trials.

**Abstract:**

Background: The current study aimed to assess the association between regular statin therapy and postoperative long-term all-cause and cancer-specific mortality following curative surgery for rectal cancer. The hypothesis was that statin exposure would be associated with better survival. Methods: Patients with stage I–III rectal cancer undergoing surgical resection with curative intent were extracted from the nationwide, prospectively collected, Swedish Colorectal Cancer Register (SCRCR) for the period from January 2007 and October 2016. Patients were defined as having ongoing statin therapy if they had filled a statin prescription within 12 months before and after surgery. Cox proportional hazards models were employed to investigate the association between statin use and postoperative five-year all-cause and cancer-specific mortality. Results: The cohort consisted of 10,743 patients who underwent a surgical resection with curative intent for rectal cancer. Twenty-six percent (*n* = 2797) were classified as having ongoing statin therapy. Statin users had a considerably decreased risk of all-cause (adjusted hazard ratio (HR) 0.66, 95% confidence interval (CI): 0.60–0.73, *p* < 0.001) and cancer-specific (adjusted HR 0.60, 95% CI: 0.47–0.75, *p* < 0.001) mortality up to five years following surgery. Conclusions: Statin use was associated with a lower risk of both all-cause and rectal cancer-specific mortality following curative surgical resections for rectal cancer. The findings should be confirmed in future prospective clinical trials.

## 1. Introduction

Rectal cancer remains one of the most common malignancies worldwide [1]. Treatment of rectal cancer often includes different neoadjuvant treatment modalities, but resection surgery remains the cornerstone of curative treatment. Notwithstanding recent advancements in perioperative care, a diagnosis of locally advanced rectal cancer confers an actuarial five-year mortality rate between 11% and 28% [2]. In the immediate postoperative period, morbidity is believed to be exacerbated by an inflammatory stress response induced by the surgical trauma, giving rise to both surgical and non-surgical complications [3,4,5,6,7].

Statins (3-hydroxy-3-methyl-glutaryl-coenzyme A (HMG-CoA) reductase inhibitors) may improve postoperative survival through mechanisms other than their widely recognized lipid-lowering actions [8]. These so-called pleiotropic effects include the reduction of inflammatory responses, as well as direct cardiovascular protection that involves improvement of endothelial function and prevention of thrombotic events [9]. Additionally, statins are suggested to possess anti-neoplastic properties that halt carcinogenic progression through mechanisms such as tumor growth suppression, angiogenesis inhibition, and induction of apoptosis [10]. Statin treatment may therefore be advantageous to patients undergoing colorectal cancer surgery in both the immediate and long-term postoperative period.

Several studies have examined the effect of statins on perioperative physiology [11,12,13,14,15,16], with some demonstrating a positive association between statin use and improved short-term postoperative outcomes [11,12]. The authors previously reported a reduction in 90-day postoperative mortality after colon or rectal cancer surgery which was independently associated with premorbid statin use [17,18]. However, the duration of the survival benefit following curative surgical resection for colorectal cancer in patients taking statins has been incompletely examined [19].

Using data from a large population-based register, we evaluated the relationship between perioperative statin use and long-term mortality after surgical resection for rectal cancer, hypothesizing that statin use is associated with better long-term survival of rectal cancer patients.

## 2. Methods

### 2.1. Study Setting

This study was carried out in accordance with the Declaration of Helsinki and STROBE guidelines (Appendix A), following approval by the Swedish Ethical Review Authority (Ref. 2018/400, Uppsala, Sweden). Clinical and demographic data for all adults (≥18 years) with stage I–III rectal cancer undergoing surgical resection with curative intent were extracted from the Swedish Colorectal Cancer Register (SCRCR) for the period from 1 January 2007 and 1 October 2016. The SCRCR is a prospectively compiled national register that contains information about patient demographics, tumor staging, surgical treatment, and neoadjuvant/adjuvant therapy [20]. In the specified study period, the SCRCR has a 99% coverage rate for all rectal cancer patients who underwent curative surgical resection [21]. The following variables were collected from the SCRCR: age, sex, tumor location, American Joint Committee on Cancer (AJCC) cTNM stage, American Society of Anesthesiologist (ASA) classification, type of surgical resection and technique (open or minimally invasive), neoadjuvant and adjuvant oncologic therapy, and dates of surgery and hospital discharge. This information was subsequently cross-referenced with data from the National Patient, Prescribed Drug, and Cause of Death registers maintained by the Swedish National Board of Health and Welfare using the patients’ unique national social security numbers. The age-adjusted Charlson Comorbidity Index (CCI) was calculated using comorbidity data from the National Patient Register [22]. An application for the retrieval of patient records was submitted to the SCRCR at the end of 2018. The complete database was received in mid-2019 after being cross-referenced with the data obtained from the Swedish Board of Health and Welfare. The primary outcome of interest was five-year all-cause mortality, and the secondary outcome was five-year rectal cancer-specific mortality.

### 2.2. Statin Therapy

Information related to statin prescriptions (ACT-code C10AA) was obtained from the Swedish Prescribed Drug Register (Swedish National Board of Health and Welfare). This register contains data on all prescribed drugs in Sweden. Patients were defined as having ongoing statin therapy if they collected a prescription for a statin within 12 months before and after surgery. Patients who filled a prescription prior to surgery but did not survive past the first postoperative year were also categorized as having ongoing statin therapy. Twelve months was chosen as a suitable cut-off given that statins are typically dispensed on an annual basis in Sweden. The postoperative duration of 12 months was chosen to confirm that it was not discontinued postoperatively. The study cohort was subsequently divided into two groups: statin users and non-users.

### 2.3. Statistical Analyses

Data regarding patient clinical characteristics and outcomes were processed using descriptive statistical methods. Results are presented as means ± standard deviations (SD) for normally distributed continuous variables, medians and quartiles for asymmetrically distributed continuous variables, or as counts and percentages for categorical variables. The statistical significance of differences between groups was determined using the Student’s t-test or Mann–Whitney U test for continuous variables, and the chi-square test for categorical variables. The relationship between statin therapy and postoperative mortality was analyzed with a Cox proportional hazards model. The regression model was adjusted for age, sex, ASA classification, CCI, cancer stage, neoadjuvant therapy, adjuvant therapy, surgical technique, type of surgery, and year of surgery. The results of the regression analyses are reported as adjusted hazard ratios (adj. HR) with 95% confidence intervals (CI). Kaplan–Meier curves were plotted to further illustrate all-cause and cancer-specific mortality up to five years after surgery. Patients who were followed for less than five years were censored from the end of their last follow-up. For cancer-specific mortality, censoring was additionally defined as death from any other cause. A two-sided *p*-value of less than 0.05 was considered statistically significant. Multiple imputations by chained equations were used to manage the missing values present in ASA classification and neoadjuvant therapy. All analyses were completed using R statistical programming language, version 4.0.5 (R Foundation for Statistical Computing, Vienna, Austria).

## 3. Results

The study cohort consisted of 10,743 patients who underwent surgical resection with curative intent for rectal cancer. Twenty-six % (*n* = 2797) of them were classified as having ongoing statin therapy. This figure was comparable to the statin use in the general Swedish population within the same age range during the study period (25%) [23]. Preoperative patient demographics and clinical characteristics are presented in Table 1. Statin users were older (71 ± 8 years vs. 68 ± 12, *p* < 0.001), more likely to be male (68.7% vs. 57.2%, *p* < 0.001), and possessed a greater surgical risk at the time of surgery based on their pre-operative ASA classification (ASA ≥ 3: 37.4% vs. 17.1%, *p* < 0.001). Lower prevalence of advanced cancer was seen in those with statin treatment (stage III: 36.3 vs. 39.0%, *p* = 0.002), and thus, this group was less subjected to neoadjuvant (8.3% vs. 13.7%, *p* < 0.001) and adjuvant therapy (16.2% vs. 20.7%, *p* < 0.001) (Table 1).

As outlined in Table 2, statin users had a significantly higher comorbidity burden. A higher fraction of patients had a CCI score ≥7 compared to non-users (25.2% vs. 11.5%, *p* < 0.001) (Table 2). Despite this, statin users experienced lower crude all-cause and cancer-specific mortality the first year after surgery. Crude all-cause mortality remained significantly reduced for each year up to three years after surgery in statin users compared to non-users (Table 3).

After adjusting for the previously mentioned confounding factors, statin use was associated with a significant improvement in survival, displaying a 34% lower risk of death from all causes up to five years after surgery (adj. HR 0.66, 95% CI: 0.60–0.73, *p* < 0.001) (Table 4, Figure 1). Additionally, compared to non-users, statin recipients also demonstrated a 40% lower risk of cancer-specific mortality up to five years after surgery (adjusted HR 0.60, 95% CI: 0.47–0.75, *p* < 0.001) (Table 4, Figure 2).

## 4. Discussion

The current study is the first to date to investigate the impact of statin therapy on long-term mortality following curative rectal cancer surgery. In this large-scale nationwide observational study involving more than 10,000 patients, having an ongoing statin prescription was associated with a lower risk of mortality after undergoing surgical resection with curative intent for AJCC Stage I–III rectal cancer. This association was observed for both all-cause and cancer-specific mortality extending up to five years after surgery.

The association between statin therapy and a reduction in all-cause mortality in patients with colorectal cancer has been previously studied; recently, the results have been aggregated in a systematic review and meta-analysis of statin use and long-term mortality, including a total of 130,994 patients, that demonstrated that statin users had improved long-term survival in terms of both all-cause and cancer-specific mortality, regardless of whether they were administered statins before or after being diagnosed with colorectal cancer. Pre-diagnosis statin treatment was associated with a 15% reduction in all-cause mortality (HR 0.85, 95% CI, 0.79–0.92) and an 18% decrease in the risk of cancer-specific mortality (HR 0.82, 95% CI, 0.79–0.86). Similarly, post-diagnosis statin use led to a 14% reduction in all-cause mortality (HR 0.86, 95% CI, 0.76–0.98) and a 21% reduction in cancer-specific mortality (HR 0.79, 95% CI, 0.70–0.89) [24]. Despite inconsistencies in prior study results, the overall pool of retrospective data suggests that statins may be advantageous for the long-term survival of patients with colorectal cancer.

We limited our analysis to all patients who underwent a curative surgical rectal resection, where perioperative variables that could potentially affect postoperative survival were controlled for through regression analysis. Additionally, the study population focused exclusively on rectal cancer, which has a higher risk of locoregional oncologic failure due to its anatomical position and which often necessitates multimodal therapy involving neoadjuvant radiotherapy or radio-chemotherapy, combined with more complex surgery. As a result, postoperative morbidity differs from colonic resections, with patients undergoing rectal cancer surgery exhibiting greater vulnerability to such adversities [14,25]. It is therefore of paramount importance to differentiate patients undergoing oncologic rectal resection from those undergoing colon resection, as both disease entities exhibit distinct characteristics. Nevertheless, the authors have previously published data on the association between statin use and postoperative short-term mortality following curative resection surgery for colon cancer, where ongoing statin therapy resulted in reduced rates of 90-day all-cause mortality (incidence risk ratio = 0.12, 95% CI: 0.09–0.16, *p* < 0.001) [17].

Few prior studies assessing the long-term effects of statin therapy have addressed patients with rectal cancer alone. In a regional Swedish population-based analysis, statin use at the time of rectal cancer diagnosis was significantly associated with improved all-cause, cancer-specific, and disease-free survival for patients aged 70 years or older [26]. Multiple retrospective cohort studies have demonstrated higher rates of pathologically complete responses to neoadjuvant chemoradiation following statin therapy, further supporting the use of statins as part of the concatenated treatment optimization of patients with rectal cancer [27,28,29]. An ongoing randomized controlled trial is investigating the potential effects of simvastatin in addition to standard chemoradiotherapy as preoperative treatment for rectal cancer [30]. However, it remains premature to recommend statin therapy for all patients diagnosed with rectal cancer at this time.

Statins may possess pleiotropic effects that improve postoperative survival through mechanisms unrelated to their lipid-altering actions. A downstream effect of statins’ inhibition of the conversion of HMG-CoA to mevalonate, the downregulation of the synthesis of isoprenoid intermediates, affects G-proteins (e.g., Rho, Rac, and Ras), which play vital roles in many intracellular signaling pathways. Statins may thereby provide direct cardiovascular protection by improving endothelial function, maintaining plaque stability, limiting blood clot formation, and lowering inflammatory responses [9]. These processes could explain why statin use is associated with fewer postoperative cardiovascular complications following noncardiac surgery [6]. Cardiovascular protection is essential given that up to 8% of all patients undergoing noncardiac surgery suffer from postoperative myocardial injury [31]. Additional pleiotropic actions of statins include anti-inflammatory effects that aid in hampering the immediate stress response and proinflammatory cytokine release caused by the surgical trauma. In a prospective trial, patients undergoing elective colon resection who were given perioperative simvastatin therapy had significantly lower plasma concentrations of proinflammatory cytokines such as tumor necrosis factor, interleukin-6, and interleukin-8 [16]. Lower circulating plasma concentrations of proinflammatory cytokines are closely correlated with decreased incidences of adverse events such as surgical site infections, anastomotic leak, sepsis and ventilator-associated respiratory complications following colorectal surgery [32,33]. Moreover, postoperative benefits of statin therapy are further highlighted with recent results showing that statin users had a lower risk of postoperative long-term adhesion-related complications after abdominal surgery [34].

Furthermore, statins are thought to exert anti-neoplastic effects that interfere with carcinogenic progression through HMG-CoA reductase-dependent and HMG-CoA reductase-independent pathways. The effects include suppression of tumor growth, inhibition of angiogenesis, and apoptosis induction [10]. Although the precise molecular mechanisms underlying the effects of statins on colorectal cancer cells remain unknown, enhanced cellular oxidative stress, endoplasmic reticulum stress and autophagy, altered expression of apoptotic and proliferative signaling mediators, and modulation of growth factors have all been implicated in experimental studies [35]. These effects could potentially explain why the current analysis discovered that statin users had a less advanced cancer stage at the time of surgery. Colorectal tumors are known to exhibit a high level of molecular heterogeneity, which may be relevant when evaluating the extent of the anti-neoplastic effects statins might possess [36]. As the current study did not capture data on molecular tumor characteristics of rectal cancer cells, the effects of statins on different molecular phenotypes of rectal cancer warrant further investigation. There is also the possibility that statin users might be more frequently monitored for comorbidities by health care and are, as a result, more likely to have an earlier disease presentation. Despite demonstrating reduced hazard ratios of rectal cancer-specific mortality among statin users, the current study is unable to assess the long-term anti-neoplastic benefits of statin medication as disease recurrence was not investigated. Currently, there is insufficient evidence that statins improve disease- and recurrence-free survival in colon or rectal cancer [37,38].

Overall, the current study observed that ongoing statin therapy at time of rectal cancer surgery resulted in a longitudinal beneficial survival effect, despite statin users being older, comprising a higher proportion of men, suffering from increased underlying comorbid conditions, and being less suitable for surgery; all of which are factors that have been substantially correlated with worse prognosis after colorectal surgery [39]. These demographic disparities are most likely the result of the primary indication of statin medication.

The present study has its limitations, mainly due to being a retrospective register-based study. For instance, there are inevitable residual confounders that could not be accounted for, one of them being cancer recurrence rates. The database used for the current study did not capture the response to neoadjuvant therapy or if statin therapy had any additional effects on the neoadjuvant therapy given. Despite the inclusion of surgical variables in the multivariate analysis, additional intraoperative variables such as intraoperative complications, blood loss, duration of surgery, and conversion rates, influencing postoperative survival were not available for assessment. Furthermore, other drugs that are routinely prescribed for similar indications as statins, such as acetylsalicylic acid, were not independently studied. Statin users may also have been more consistently followed up by their primary care physician for comorbidities, increasing the chance that they are referred for a colonoscopy; this would result in an earlier disease presentation, giving rise to a lead-time bias. Finally, the results do not carefully consider different doses or classes of statins. However, there is currently no evidence that the pleiotropic effects of statins differ among their various forms [40]. The strengths of this study include the utilization of a nearly complete nationwide register providing a vast dataset accessible for analysis. Only including patients who underwent surgery for rectal cancer resulted in a more homogenous patient population in terms of presenting disease and interventions, which gives similar postoperative risks in terms of prognosis. Lastly, it is unlikely that treatment management differed between patients within the cohort, as Sweden offers universal health care to its citizens, minimizing the effects of socioeconomic factors.

## 5. Conclusions

In this nationwide Swedish observational cohort study, statin therapy usage in patients undergoing curative surgical resection for rectal cancer was associated with a reduced risk of both all-cause and rectal cancer-specific mortality. While these findings are promising, they should be confirmed in future large, randomized clinical trials.

## Figures and Tables

**Figure 1 cancers-13-04288-f001:**
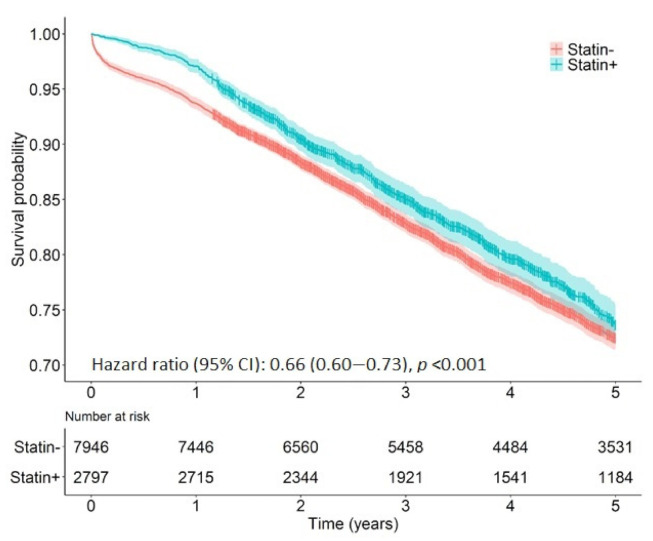
Kaplan–Meier plot of 5-year all-cause mortality for statin users and non-users following surgical resection for rectal cancer.

**Figure 2 cancers-13-04288-f002:**
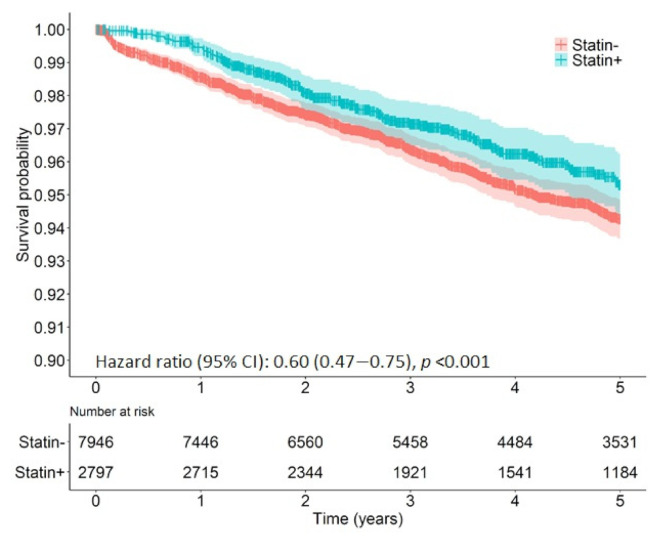
Kaplan–Meier plot of 5-year cancer-specific mortality for statin users and non-users following surgical resection for rectal cancer.

**Table 1 cancers-13-04288-t001:** Patient demographics and clinical features on the day of surgery.

Variable	Statin− (*N* = 7946)	Statin+ (*N* = 2797)	*p*-Value
Age, mean (±SD)	67.7 (±11.7)	71.2 (±8.2)	<0.001
Sex, *n* (%)			<0.001
Male	4549 (57.2)	1921 (68.7)	
Female	3397 (42.8)	876 (31.3)	
ASA classification, *n* (%)			<0.001
1	2113 (26.6)	139 (5.0)	
2	4351 (54.8)	1580 (56.5)	
3	1303 (16.4)	993 (35.5)	
4	58 (0.7)	53 (1.9)	
Missing	121 (1.5)	32 (1.1)	
Cancer stage, *n* (%)			0.002
I	2300 (28.9)	907 (32.4)	
II	2546 (32.0)	875 (31.3)	
III	3100 (39.0)	1015 (36.3)	
Neoadjuvant therapy, *n* (%)	1092 (13.7)	232 (8.3)	<0.001
Missing	2873 (36.2)	1113 (39.8)	
Adjuvant therapy, *n* (%)	1644 (20.7)	452 (16.2)	<0.001
Surgical technique, *n* (%)			0.099
Open surgery	6682 (84.1)	2314 (82.7)	
Laparoscopic surgery	1264 (15.9)	483 (17.3)	
Type of surgery, *n* (%)			0.040
Anterior resection	4175 (52.5)	1435 (51.3)	
Abdominoperineal excision	2900 (36.5)	1006 (36.0)	
Hartmann’s operation	871 (11.0)	356 (12.7)	

SD, standard deviation; ASA, American Society of Anesthesiologists.

**Table 2 cancers-13-04288-t002:** Patient comorbidities as recorded on the day of surgery.

Variable	Statin− (*N* = 7946)	Statin+ (*N* = 2797)	*p*-Value
Arrhythmia, *n* (%)	644 (8.1)	401 (14.3)	<0.001
Hypertension, *n* (%)	1469 (18.5)	1261 (45.1)	<0.001
Myocardial infarction, *n* (%)	136 (1.7)	443 (15.8)	<0.001
Congestive heart failure, *n* (%)	240 (3.0)	198 (7.1)	<0.001
Peripheral vascular disease, *n* (%)	132 (1.7)	186 (6.6)	<0.001
Cerebrovascular disease, *n* (%)	274 (3.4)	342 (12.2)	<0.001
Dementia, *n* (%)	85 (1.1)	29 (1.0)	0.969
COPD, *n* (%)	337 (4.2)	181 (6.5)	<0.001
Connective tissue disease, *n* (%)	131 (1.6)	62 (2.2)	0.063
Peptic ulcer disease, *n* (%)	122 (1.5)	60 (2.1)	0.039
Liver disease, *n* (%)	56 (0.7)	14 (0.5)	0.309
Diabetes, *n* (%)	394 (5.0)	661 (23.6)	<0.001
Hemiplegia, *n* (%)	36 (0.5)	36 (1.3)	<0.001
Chronic kidney disease, *n* (%)	83 (1.0)	70 (2.5)	<0.001
Charlson comorbidity index, *n* (%)			<0.001
≤4	3785 (47.6)	646 (23.1)	
5–6	3251 (40.9)	1447 (51.7)	
≥7	910 (11.5)	704 (25.2)	

COPD, chronic obstructive pulmonary disease.

**Table 3 cancers-13-04288-t003:** Crude postoperative outcomes.

Variable	Statin− (*N* = 7946)	Statin+ (*N* = 2797)	*p*-Value
Length of stay (days), median [IQR]	9.0 [7.0–14]	10 [7.0–15]	0.002
Missing, *n* (%)	67 (0.8)	25 (0.9)	
1-year mortality, *n* (%)			
All-cause	500 (6.3)	82 (2.9)	<0.001
Cancer-specific	111 (1.4)	15 (0.5)	<0.001
	Statin− (*N* = 7474)	Statin+ (*N* = 2606)	
2-year mortality, *n* (%)			
All-cause	917 (12.3)	262 (10.1)	0.003
Cancer-specific	194 (2.6)	51 (2.0)	0.080
	Statin− (*N* = 6756)	Statin+ (*N* = 2315)	
3-year mortality, *n* (%)			
All-cause	1304 (19.3)	394 (17.0)	0.016
Cancer-specific	259 (3.8)	71 (3.1)	0.102
	Statin− (*N* = 6119)	Statin+ (*N* = 2048)	
4-year mortality, *n* (%)			
All-cause	1635 (26.7)	507 (24.8)	0.085
Cancer-specific	319 (5.2)	87 (4.2)	0.093
	Statin− (*N* = 5430)	Statin+ (*N* = 1796)	
5-year mortality, *n* (%)			
All-cause	1899 (35.0)	612 (34.1)	0.507
Cancer-specific	357 (6.6)	100 (5.6)	0.143

**Table 4 cancers-13-04288-t004:** Hazard Ratio for 5-year all-cause and colorectal cancer-specific mortality after elective rectal cancer surgery.

Variable	All-Cause Mortality		Cancer-Specific Mortality	
	HR (95% CI)	*p*-Value	HR (95% CI)	*p*-Value
Statin therapy				
No	ref.		ref.	
Yes	0.66 (0.60–0.73)	<0.001	0.60 (0.47–0.75)	<0.001
Age	1.03 (1.02–1.04)	<0.001	1.04 (1.02–1.05)	<0.001
Sex				
Male	ref.		ref.	
Female	0.81 (0.74–0.88)	<0.001	1.05 (0.87–1.26)	0.633
ASA classification				
1	ref.		ref.	
2	1.29 (1.12–1.47)	<0.001	1.28 (0.93–1.76)	0.136
3	2.07 (1.78–2.41)	<0.001	2.38 (1.67–3.38)	<0.001
4	3.01 (2.23–4.05)	<0.001	3.33 (1.64–6.76)	<0.001
Charlson Comorbidity Index				
≤4	ref.		ref.	
5–6	1.37 (1.20–1.57)	<0.001	1.48 (1.07–2.04)	0.018
≥7	2.04 (1.74–2.40)	<0.001	2.06 (1.40–3.02)	<0.001
Cancer stage				
I	ref.		ref.	
II	1.42 (1.26–1.59)	<0.001	1.52 (1.16–2.00)	0.003
III	2.47 (2.21–2.75)	<0.001	2.51 (1.93–3.26)	<0.001
Neoadjuvant therapy				
No	ref.		ref.	
Yes	1.36 (1.16–1.60)	<0.001	1.77 (1.31–2.39)	<0.001
Adjuvant therapy				
No	ref.		ref.	
Yes	0.93 (0.82–1.05)	0.221	0.93 (0.70–1.25)	0.646
Surgical technique				
Open surgery	ref.		ref.	
Laparoscopic surgery	0.91 (0.79–1.04)	0.157	0.93 (0.67–1.28)	0.651
Type of surgery				
Anterior resection	ref.		ref.	
Abdominoperineal excision	1.53 (1.40–1.68)	<0.001	1.72 (1.39–2.14)	<0.001
Hartmann’s operation	1.77 (1.58–1.98)	<0.001	1.84 (1.40–2.42)	<0.001
Surgery year				
2007	ref.		ref.	
2008	0.90 (0.77–1.05)	0.170	0.85 (0.60–1.20)	0.349
2009	0.90 (0.77–1.04)	0.158	0.87 (0.62–1.22)	0.420
2010	0.78 (0.67–0.92)	0.003	0.66 (0.46–0.96)	0.029
2011	0.80 (0.68–0.94)	0.006	0.85 (0.60–1.20)	0.347
2012	0.67 (0.57–0.79)	<0.001	0.53 (0.36–0.77)	0.001
2013	0.68 (0.57–0.81)	<0.001	0.52 (0.34–0.78)	0.002
2014	0.67 (0.56–0.81)	<0.001	0.49 (0.32–0.77)	0.002
2015	0.68 (0.55–0.83)	<0.001	0.35 (0.20–0.61)	<0.001
2016	0.62 (0.44–0.87)	0.005	0.36 (0.15–0.84)	0.019

Cox regression model adjusted for age, sex, ASA classification, comorbidities, cancer stage, neoadjuvant therapy, adjuvant therapy, surgical technique, and type of surgery. Multiple imputation with chained equations was used to manage missing values. HR, Hazard Ratio; ASA, American Society of Anesthesiologists.

## Data Availability

The data presented in this study are available on request from the corresponding author after additional ethical approval is granted by the National Ethical Authority.

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
