# Peer review of "Statin Use and Long-Term Mortality after Rectal Cancer Surgery"

_cancers, 2021, doi:10.3390/cancers13174288_

Round 1

Reviewer 1 Report

The manuscript with ID cancers-1282200, entitled „Statin Use and Long-term Mortality after Rectal Cancer Surgery“ represent an interesting study, conducted on a large set of rectal cancer patients. The power of this epidemiological study is such that it merits publication in Cancers.

There are some points and questions that need to be addressed. An additional information has to be added on downstaging of the patients following neoadjuvant chemoradiotherapy. If the patients underwent surgical resection, does it mean that they responded to neoadjuvant therapy poorly?

Some effort should be dedicated to the comparison of prognostic value of statin ise in colon and rectal cancers (Discussion).

Regarding the mode of action: Could statin modulate oxidative stress, arising oxidative DNA damage and its repair? This would be of importance in relation to inflammatory processed in colorectal carcinogenesis. Some aspects have been reviewed by Murphy et al., Mol Aspects Medicine 2019.

How is the situation in RC patients with advanced disease (i.e. TNM IV)?

Please comment more the differences in age and sex regarding statin treatment. The same applies for ASA and TNM distribution.

Discussion: There is also a huge heterogeneity, both molecular and pathological, in relation to e.g. tumor location. Could these different molecular features (phenotypes) affect the benefit from statin use?

L.59: please add: …long-term survival of rectal cancer patients.

L.98: …were processed perhaps.

L.199: Twenty six % of them...

Author Response

Response to Reviewers’ comments

Reviewer #1

The manuscript with ID cancers-1282200, entitled “Statin Use and Long-term Mortality after Rectal Cancer Surgery“ represent an interesting study, conducted on a large set of rectal cancer patients. The power of this epidemiological study is such that it merits publication in Cancers.

There are some points and questions that need to be addressed. An additional information has to be added on downstaging of the patients following neoadjuvant chemoradiotherapy. If the patients underwent surgical resection, does it mean that they responded to neoadjuvant therapy poorly?

Response: We thank the reviewer for their critical review of our manuscript. The question raised by the reviewer is a valid one, however, due to the retrospective nature of the current study the authors cannot make any comment on the response to neoadjuvant therapy. This limitation has now been mentioned in the discussion section of the text:

“The database used for the current study did not capture the response to neoadjuvant therapy or if statin therapy had any additional effects on the neoadjuvant therapy given.”

Some effort should be dedicated to the comparison of prognostic value of statin use in colon and rectal cancers (Discussion).

Response: Thank you for this comment. The authors have previously investigated the association between the preoperative statin use and short-term (90-day) mortality in patients subjected to elective curative colon cancer surgery. We have also another study under review investigating the long-term mortality after elective curative colon cancer surgery where we see a 5-year beneficial effect in both all-cause (Ajd HR 0.70 (95% CI: 0.65-0.75), p < 0.001) and cancer specific (Ajd HR 0.75 (95% CI: 0.65-0.85), p < 0.001) mortality. As mentioned in the discussion section, the authors opted to investigate the statin use and its association with postoperative mortality in oncological colon and rectal cancer resections separately, since both disease entities exhibit distinct characteristics with different postoperative mortality risks.

We have extended the discussion text per the reviewer’s request to include the data from colon cancer surgery as well:

“Nevertheless, the authors have previously published data on the association between statin use and postoperative short-term mortality following curative resection surgery for colon cancer, where ongoing statin therapy resulted in reduced rates of 90-day all-cause mortality (incidence risk ratio = 0.12, 95% CI: 0.09-0.16, p < 0.001).17

Regarding the mode of action: Could statin modulate oxidative stress, arising oxidative DNA damage and its repair? This would be of importance in relation to inflammatory processed in colorectal carcinogenesis. Some aspects have been reviewed by Murphy et al., Mol Aspects Medicine 2019.

Response:  Thank you for highlighting this important information, i.e. action of statins and their role in modulating oxidative stress. This piece of information is now added in the discussion and referenced:

 “Although the precise molecular mechanisms underlying the effects of statins on colorectal cancer cells remain unknown, enhanced cellular oxidative stress, endoplasmic reticulum stress and autophagy, altered expression of apoptotic and proliferative signaling mediators, and modulation of growth factors have all been implicated in experimental studies.35

How is the situation in RC patients with advanced disease (i.e. TNM IV)?

Response: Thank you for this question. The authors deliberately opted not to include this patient population due to the high mortality rates, as well as some of these patients will have other surgeries, for example hepatectomies, due to metastasis which would need further in depth analysis.

Please comment more the differences in age and sex regarding statin treatment. The same applies for ASA and TNM distribution.

Response: Thank you for this comment. Not surprisingly, statin users are older with higher surgical risk (based on their preoperative ASA classification) which to some extent reflects their comorbidly burden. Further, hypercholesterolemia and cardiovascular diseases are still more often diagnosed in the male population, and hence, statins are more often prescribed in this patient cohort. Previous studies, referenced in the manuscript, have shown that these factors bring increased risks of adverse effects after colorectal cancer surgery.  

 The following section have now been added to the discussion:

 “ Overall, the current study observed that ongoing statin therapy at time of rectal cancer surgery resulted in a longitudinal beneficial survival effect, despite statin users being older, constituting of a higher proportion of men, suffer from increased underlying comorbid conditions, and are less suitable for surgery; all of which are factors that have been substantially correlated with worse prognosis after colorectal surgery.39 These demographic disparities are most likely the result of the primary indication of statin medication.”

Discussion: There is also a huge heterogeneity, both molecular and pathological, in relation to e.g. tumor location. Could these different molecular features (phenotypes) affect the benefit from statin use?

Response: This could well be the case, however, all patients in the current study underwent surgery with curative intent. Due to the retrospective nature of this study the authors cannot make any comments on the effect of statins on different rectal cancer phenotypes. We have added this limitation to the discussion section of the manuscript.

 “Colorectal tumors are known to exhibit a high level of molecular heterogeneity, which may be relevant when evaluating the extent of the anti-neoplastic effects statins might posess.36 As the current study did not capture data on molecular tumor characteristics of rectal cancer cells, the effects of statins on different molecular phenotypes of rectal cancer warrant further investigation.”

 L.59: please add: …long-term survival of rectal cancer patients.

L.98: …were processed perhaps.

L.199: Twenty six % of them...

Response: Thank you for bringing these sentences to our attention which now has been corrected.

Reviewer 2 Report

Well written paper, clearly describing the methods and statistics for this retrospective registry study. The results are clearly presented in table and figure format and the conclusions are appropriate. Findings and limitations are discussed in context of existing literature and future recommendations clear. Finally, results of this study are clinically relevant.

The following additional points can be added regarding the Discussion section:

1. Adding the number of patients included in the meta-analysis (line 171, ref 25) would help contextualize the manuscript within existing literature

2. Data on the use of neoadjuvant therapy is missing for a large proportion of the patients (more than 30%). Given the importance of neoadjuvant therapy in rectal cancer this should be included as a limitation of the study and potential impact of this missing data included in the discussion.

Author Response

Response to Reviewers’ comments

Reviewer #2

Well written paper, clearly describing the methods and statistics for this retrospective registry study. The results are clearly presented in table and figure format and the conclusions are appropriate. Findings and limitations are discussed in context of existing literature and future recommendations clear. Finally, results of this study are clinically relevant.

The following additional points can be added regarding the Discussion section:

  1. Adding the number of patients included in the meta-analysis (line 171, ref 25) would help contextualize the manuscript within existing literature

Response: Thank you for this suggestion. Per the reviewer’s suggestion these has been added in the manuscript.

  1. Data on the use of neoadjuvant therapy is missing for a large proportion of the patients (more than 30%). Given the importance of neoadjuvant therapy in rectal cancer this should be included as a limitation of the study and potential impact of this missing data included in the discussion.

Response: Thank you for this valid and important comment. All missing values were handled by multiple imputation and is mentioned under the statistical analyses section:

Multiple imputations by chained equations were used to manage the missing values present in ASA classification and neoadjuvant therapy.”

Reviewer 3 Report

The usage and effect of statins on postoperative results for rectal cancer is interesting point and this could be object of future studies. My only concern is practical point and interpretation of statistics. The positive thing of this study is large number of patients, but from other hand when you have thousands of patients, you get statistically significant difference between the groups even if these differences are very small. As you see in table 3, differences between the groups in survival are very small (1-2%), and this is because of different numbers of advanced cancer in groups as mentioned in article. Practical point of this is that to get 1% survival benefit 99% of patients will not have survival benefit from statins. The biggest shortage of this paper is that there is no  data about recurrence rates, this could also give additional information. 

Author Response

Response to Reviewers’ comments

Reviewer #3

The usage and effect of statins on postoperative results for rectal cancer is interesting point and this could be object of future studies. My only concern is practical point and interpretation of statistics. The positive thing of this study is large number of patients, but from other hand when you have thousands of patients, you get statistically significant difference between the groups even if these differences are very small. As you see in table 3, differences between the groups in survival are very small (1-2%), and this is because of different numbers of advanced cancer in groups as mentioned in article. Practical point of this is that to get 1% survival benefit 99% of patients will not have survival benefit from statins.

Response: The authors thank the reviewer for their time to critically review our work. The reviewer is correct in their assessment that the crude mortality, not considering differences in age, comorbidities, type of surgery and approach etc., does only difference up to 2% for each postoperative year up to 5-years. It is also important to emphasize that this is in patients who had surgery with curative intent and the postoperative mortality rate is low in these circumstances. Nevertheless the 5-year mortality risk, adjusted hazard ratio for all cause and cancer specific mortality is 0.66 and 0.60, respectively.

The biggest shortage of this paper is that there is no  data about recurrence rates, this could also give additional information.

Response: The authors are in agreement with the reviewer that recurrence rates would have given additional information, regrettably this information is not captured in the dataset used for this study. We have added this limitation in the discussion section of the manuscript:

 ”The present study has its limitations, mainly due to being a retrospective register-based study. For instance, there are inevitable residual confounders that could not be accounted for, one of them being cancer recurrence rates.”
